# Quantification of differential gene expression by multiplexed targeted resequencing of cDNA

Peer Arts[1,*], Jori van der Raadt[1,2,*], Sebastianus H.C. van Gestel[1,2,*], Marloes Steehouwer[1], Jay Shendure[3,4], Alexander Hoischen[1,5,**] & Cornelis A. Albers[1,2,**]

Whole-transcriptome or RNA sequencing (RNA-Seq) is a powerful and versatile tool for functional analysis of different types of RNA molecules, but sample reagent and sequencing cost can be prohibitive for hypothesis-driven studies where the aim is to quantify differential expression of a limited number of genes. Here we present an approach for quantification of differential mRNA expression by targeted resequencing of complementary DNA using single-molecule molecular inversion probes (cDNA-smMIPs) that enable highly multiplexed resequencing of cDNA target regions of ~100 nucleotides and counting of individual molecules. We show that accurate estimates of differential expression can be obtained from molecule counts for hundreds of smMIPs per reaction and that smMIPs are also suitable for quantification of relative gene expression and allele-specific expression. Compared with low-coverage RNA-Seq and a hybridization-based targeted RNA-Seq method, cDNA-smMIPs are a cost-effective high-throughput tool for hypothesis-driven expression analysis in large numbers of genes (10 to 500) and samples (hundreds to thousands).

[1] Department of Human Genetics, Donders Institute for Brain, Cognition and Behaviour, Radboud University Medical Center, PO Box 9101, 6500 HB, Nijmegen, The Netherlands. [2] Department of Molecular Developmental Biology, Radboud Institute for Molecular Life Sciences, Radboud University, PO Box 9101, 6500 HB, Nijmegen, The Netherlands. [3] Department of Genome Sciences, University of Washington, Foege Building S-250, Box 355065, 3720 15th Ave NE, Seattle, Washington 98195-5065, USA. [4] Howard Hughes Medical Institute, Seattle, Washington 98195, USA. [5] Department of Internal Medicine and Radboud Center for Infectious Diseases (RCI), Radboud University Medical Center, PO Box 9101, 6500 HB, Nijmegen, The Netherlands. * These authors contributed equally to this work. ** These authors jointly supervised this work. Correspondence and requests for materials should be addressed to C.A.A. (email: kees.albers@radboudumc.nl).

Whole-transcriptome or RNA sequencing (RNA-Seq) is a powerful and versatile tool for functional analysis of different types of RNA molecules in a wide variety of applications, ranging from fundamental cell biology to clinical studies of the consequences of genomic variation and environmental perturbations[1,2]. Besides genome-wide differential expression analysis, RNA-seq can be used to quantify allele-specific expression, alternative splicing or gene fusions events[1,2]. A limitation of whole-transcriptome sequencing is that the per-sample reagent and sequencing cost can be prohibitive for hypothesis-driven studies where the aim is to quantify differential expression of a limited set of genes in a large number of experimental conditions or samples. Alternatively, quantitative PCR (qPCR) is a cost-effective and robust technique to assay a small number of genes in a medium number of samples. However, qPCR rapidly becomes labour intensive and expensive as the number of samples and genes increases. Targeted resequencing approaches have the potential to close the gap between whole-transcriptome sequencing and qPCR. Hybridization-based capture approaches using biotin-labelled oligonucleotide RNA or DNA probes have been used to better characterize splicing or fusions of lowly expressed transcripts[3–5]. While significantly reducing the required total number of sequencing reads, these approaches still have considerable per-sample reagent costs, limiting scaling up to large numbers of samples. Other approaches such as Luminex xMAP handle thousands of samples for $\sim 1,000$ genes[6,7] but require specialized equipment.

Single-molecule molecular inversion probes (smMIPs)[8] may fill the gap between qPCR and RNA-Seq as it allows a library-free enrichment (that is, enrichment and library preparation in a single step), a high degree of multiplexing of both targets and samples, single-molecule counting via degenerate tags and the protocol to generate the sequencing library can be performed in a lab with standard PCR equipment. The cost of a single smMIP at $\sim 7$ USD[8] is similar to that of a qPCR primer pair; one column-based synthesis of smMIPs (25 nmol scale) is sufficient for millions of independent reactions. smMIPs were previously developed as a method for genotyping[9,10] and targeted DNA resequencing to identify rare genetic variation in tens to hundreds of genes in thousands of individuals[8] or estimate genomic copy number variation[11]. Circular padlock probes[10], which laid the basis for smMIPs, have been used for estimation of allelic ratios in complementary DNA (cDNA), but target only 1 nucleotide of sequence and do not allow for single-molecule counting[12].

Here we show that smMIPs can be applied to cDNA to provide accurate estimates of differential expression, and that they are also suitable for quantification of relative gene expression and allele-specific gene expression. We compare the performance of cDNA-smMIPs to that of CaptureSeq and low-coverage RNA-Seq for targeted gene expression studies. Finally, we show that cDNA-smMIPs are cost effective compared with alternative approaches.

## Results

**Outline of method**. We developed an experimental approach and dedicated statistical model (Fig. 1a) to quantify differential expression, relative expression and allelic ratios with molecule counts from cDNA-smMIPs. Our approach consists of applying single-molecule molecular inversion probes to cDNA (reverse transcribed RNA). The protocol is similar to that for MIP or smMIP-based resequencing of DNA[8,13]; the key differences are in the design of smMIPs and in the overall ratio of cDNA molecules to smMIPs that we increased 10-fold compare to genomic DNA (gDNA) to account for the large dynamic range of transcript abundance (see Methods, 'smMIP capture'). We designed between 5 and 10 cDNA-specific smMIPs per gene, of which some span exon–exon boundaries and some fall inside exons. We used individual molecule counts obtained from the unique molecular identifiers (UMIs) in the smMIPs to quantify expression with a Bayesian statistical model.

**Comparison with external RNA Controls (ERCC)**. To determine the accuracy of expression quantification with cDNA-smMIPs, we used artificial transcripts with precisely known concentrations (External RNA Control Consortium (ERCC)[14]. There are two mixes (ERCC1 and ERCC2) containing the same 92 ERCC transcripts. These transcripts have different concentrations in each mix: they are divided into four groups such that the ratio of concentrations ERCC1 versus ERCC2 transcripts is respectively $0.5 \times$, $0.67 \times$, $1.0 \times$ (no difference) and $4.0 \times$. We added ERCC1[2] RNA to total RNA from human peripheral blood mononuclear cells (PBMCs), and generated cDNA from this combined artificial and human RNA sample, yielding one ERCC1/PBMC cDNA sample and one ERCC2/PBMC cDNA sample. Four smMIP captures per cDNA sample were performed (two with 10 ng cDNA input and two with 50 ng cDNA input) with 337 smMIPs targeting the 92 ERCC transcripts (5–9 smMIPs/transcript).

In keeping with results from smMIP-based resequencing of DNA, there was considerable variation between probes targeting the same cDNA transcript (Supplementary Fig. 1 and Supplementary Table 1). However, we found that correlation with the known concentrations was high when we averaged the expression as estimated by multiple probes targeting the same transcript ($R^2 = 0.91 \pm 0.02$, s.d. from 8 technical replicates, Supplementary Fig. 2). The transcript-level correlation was similar to that of the targeted RNA-Seq method CaptureSeq[5] that uses hybridization of biotin-labelled oligonucleotide probes to enrich for selected transcripts (Supplementary Fig. 3).

**Accuracy of differential expression estimates**. We evaluated the accuracy of differential expression (DE) estimates. Variability in capture efficiency between smMIPs should not affect DE estimates because the abundance estimated by the same probe is compared between conditions. However, we noted that the difference in expression values between two replicates for ERCC1 was correlated with the difference between two replicates for ERCC2, indicating the presence of a systematic probe bias that is independent of the experimental condition (Supplementary Fig. 4). We developed a Bayesian hierarchical model to estimate differential expression while correcting for this bias (see Methods). The model estimates a single normalized expression value for each probe and condition from all replicates for a given condition.

Our approach yielded accurate estimates of differential expression from individual cDNA-smMIP probes by combining counts from four technical replicates (Fig. 1b). We next averaged the differential expression estimates of probes targeting the same transcript to obtain a transcript-level estimate of differential expression. Also, at the transcript level the accuracy (difference between expected and observed fold change) of cDNA-smMIPs was high, successfully distinguishing between 0.67- and 0.5-fold changes (Fig. 1c, Supplementary Tables 1 and 2). Accuracy of CaptureSeq (based on, respectively, 4 and 5 replicates for ERCC1 and ERCC2) was lower that underestimated the fold change for transcripts with $4 \times$ difference in abundance. However, the precision (variation in estimated fold change within each group) of CaptureSeq was somewhat higher than that of cDNA-smMIPs (see Supplementary Figs 5 and 6 for comparisons between

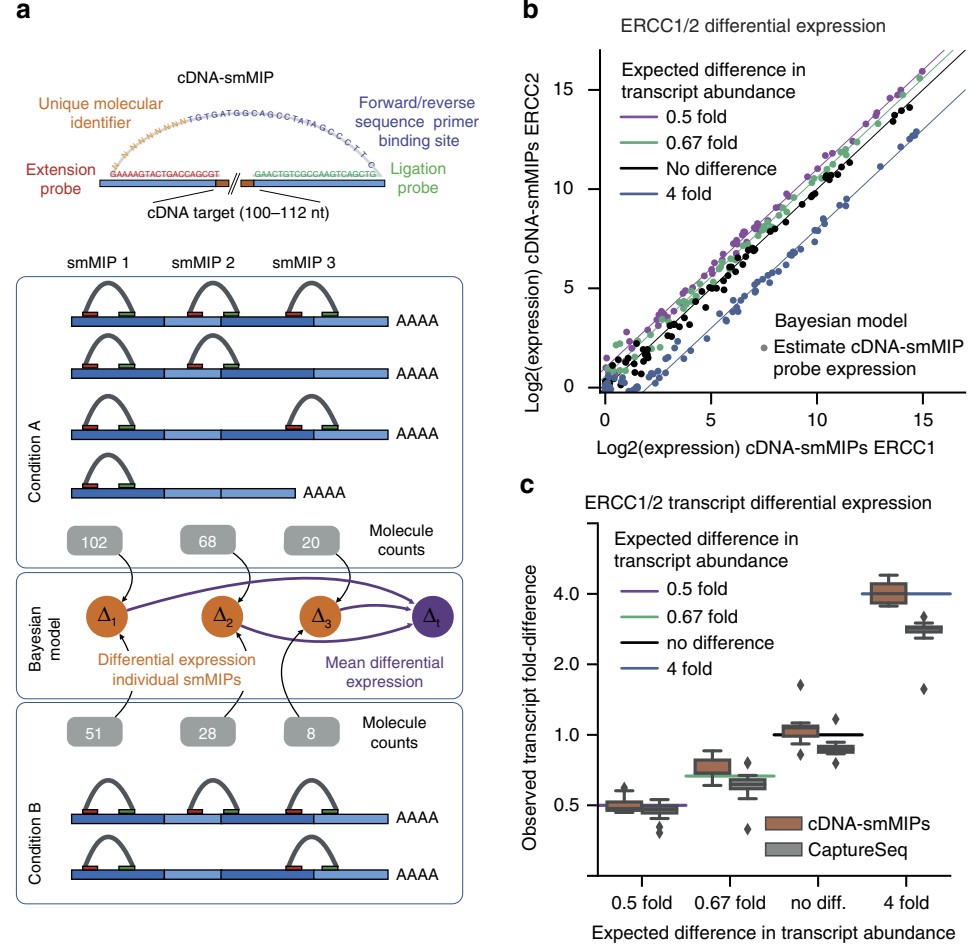

**Figure 1 | Evaluation of cDNA-smMIPs for estimation of differential expression with artificial transcripts.** (**a**) Outline of the approach. (**b**) Accuracy of differential expression estimates from 337 individual cDNA-smMIPs targeting 92 ERCC transcripts in condition ERCC1 and ERCC2. The 92 transcripts are divided into four groups; for each group the difference in transcript abundance between condition ERCC1 and ERCC2 is known, and is indicated by the solid lines. Four technical capture replicates were performed on respectively one cDNA sample for condition ERCC1 and one cDNA sample for condition ERCC2. The expression value for each smMIP is estimated using a Bayesian model. Data points are coloured according to their expected expression fold difference. (**c**) Comparison of differential expression quantification by cDNA-smMIPs and the previously published method CaptureSeq[5] that performs targeted RNA-Seq by biotin-labelled oligonucleotide hybridization. Only transcripts with log2-expression values of > 4 were included for both methods. Differential expression estimated with the Bayesian model from respectively four ERCC1 and ERCC2 cDNA-smMIPs capture replicates is compared with differential expression estimates from respectively 4 ERCC1 and 5 ERCC2 replicates for CaptureSeq (see respectively Supplementary Tables 1 and 2 for statistics).

individual replicates for respectively cDNA-smMIPs and CaptureSeq). Thus, accurate quantification of relative gene expression, which involves comparisons of different genes in the same condition, requires multiple smMIPs per transcript; in contrast, accurate quantification of differential expression, which involves comparing the same gene in different conditions, can be achieved with a single smMIP, but can be further improved by combining estimates from multiple smMIPs.

To gain more insight into the relative performance of CaptureSeq and cDNA-smMIPs, we compared sensitivity of transcript detection. CaptureSeq detected low-abundance transcripts with higher sensitivity (Supplementary Fig. 7a). It has previously been observed that CaptureSeq read counts saturate for the high-abundance ERCC transcripts and that consequently CaptureSeq read count is not linearly correlated with transcript abundance (see Supplementary Figs 3 and 5 in ref. 5). This effectively increases sensitivity to low-abundance transcripts. In contrast, cDNA-smMIPs molecule counts were linearly correlated with ERCC transcript abundance (Supplementary Figs 8 and 9); thus, more reads are accounted for by the high-abundance transcripts, and cDNA-smMIPs

detected overall fewer transcripts than CaptureSeq. However, on the subset of ERCC transcripts whose concentration was in the range where CaptureSeq quantification is linear, detection sensitivity of cDNA-smMIPs and CaptureSeq was similar (Supplementary Fig. 7b).

**Evaluation using endogenous transcripts.** We next evaluated the performance of cDNA-smMIPs on endogenous transcripts of Epstein–Barr transformed lymphoblast cell lines (EBVs). We used RNA-Seq data for EBV cell lines of 660 samples from the Geuvadis project[15] to design smMIPs for the most highly expressed transcript of 12 genes (95 smMIPs in total) that spanned a range of expression values (Supplementary Table 3). To evaluate reproducibility, cDNA-smMIP capture experiments were performed for two different cell lines originating from two individuals (EBV2 and EBV3) in two separate experiments. In the second experiment, two experimenters independently performed smMIP capture using the same cDNA sample as input. In the second experiment, cDNA was created from the same RNA stock as the first experiment. Again, four technical capture replicates (two with 10 ng cDNA input and two with 50 ng cDNA input)

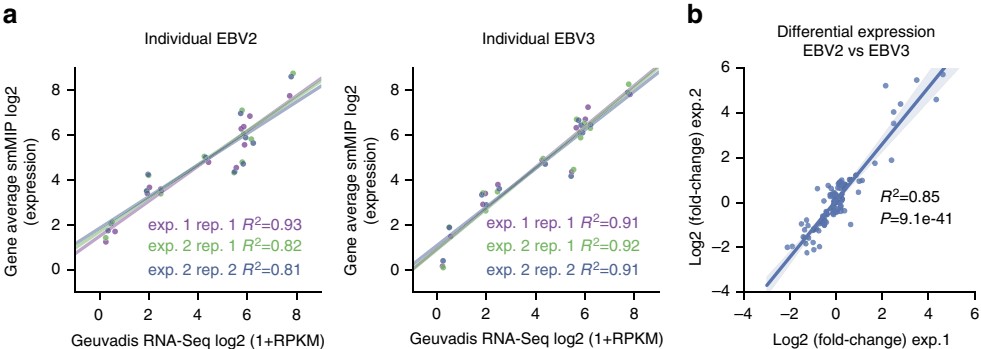

**Figure 2 | Validation of cDNA-smMIPs using on lymphoblastoid cell lines.** (**a**) Quantification of relative gene expresssion on endogeneous transcripts of EBV-transformed lymphoblastoid cell lines compared with average gene expression from RNA-Seq data of 660 samples in the Geuvadis project[15] for the same cell type. Two experiments were performed (exp. 1 and exp. 2); in the second experiment, two technical replicates were created by two independent experimenters (designated by Rep. 1 and Rep. 2, see Supplementary Table 2). cDNA for experiments 1 and 2 was generated independently from the same RNA for EBV2 and EBV3. (**b**) Concordance of differential expression between sample EBV2 and EBV3 for individual smMIPs ($N = 95$).

per cell line were used in each experiment. We then applied our Bayesian model to estimate normalized smMIP expression values for each condition and averaged these to obtain gene-level estimates of expression; the gene average was compared with the population average of $\log2(1 + RPKM)$ gene expression values estimated by the Geuvadis project. cDNA-smMIPs performed well, with Pearson's correlation $R^2$ between cDNA-smMIPs and RNA-Seq estimates of gene expression ranging from 0.81 to 0.92 (Fig. 2a and Supplementary Table 4). We observed a slight discrepancy for the EBV2 cell line, where the correlation was respectively $R^2 = 0.91$ and $R^2 = 0.81$ in the first and second experiments. Interestingly, in the second experiment the concordance between the two independent experimenters was very high for this sample ($R^2 = 0.998$, $P = 1e - 14$; two-sided Pearson's correlation test), suggesting variability in cDNA synthesis as a potential cause for the discrepancy. The DE estimates for the two cell lines were reproducible between the first and second experiments ($R^2 = 0.85$, $P = 9e - 41$; two-sided Pearson's correlation test. Fig. 2b).

**Comparison with low-coverage whole-transcriptome RNA-Seq.** We compared the performance of cDNA-smMIPs to unbiased low-coverage whole-transcriptome RNA sequencing at the same number of sequencing reads. The expectation is that enrichment with cDNA-smMIPs results in increased sequencing depth at the transcripts of interest. Indeed, for our panel of 12 genes targeted with 95 cDNA-smMIPs, molecule counts per gene were ∼100-fold higher than fragment counts (read pairs) obtained with RNA-Seq (Fig. 3a) at the same total number of reads. As a result, the number of genes for which expression was detected was increased (Fig. 3b) and reproducibility of estimated fold changes was higher (Fig. 3c and Supplementary Fig. 11) for cDNA-smMIPs. cDNA-smMIPs make it possible to obtain molecule counts for specific exons or exon–exon boundaries. As expected, the molecule counts of cDNA-smMIPs individual target regions are also similarly ∼100-fold higher than the fragment count of whole-transcriptome RNA-Seq in the smMIP target regions (Fig. 3d). This facilitates expression analysis of specific isoforms.

**Comparison with RNA-Seq in biological application.** We next sought to replicate with cDNA-smMIPs our previous study[16] where we used RNA-Seq to characterize the immune response in human PBMCs following *in vitro* stimulation with heat-killed (HK) *Candida albicans*, a common cause of fungal infections. We obtained PBMCs from a new anonymous blood donor and

added HK *C. albicans* to the PBMC culture medium for a period of 24 h as described previously[16] (Fig. 4). We used data from two smMIP capture replicates per condition (Supplementary Table 5). We then estimated differential expression between the stimulated PBMCs and the control condition without *C. albicans*. To compare with the RNA-Seq estimate, probe-level estimates of differential expression from the Bayesian model were averaged to obtain gene-level estimates and associated confidence intervals. We were able to replicate with cDNA-smMIPs our previous finding[16] that mRNA levels of the interferon-γ response genes *IFIT1* and *IFNG* are strongly upregulated following *C. albicans* stimulation (Fig. 4 and Supplementary Fig. 11). Thus, cDNA-smMIPs can be applied to primary cells to study the molecular mechanisms in biological systems.

**Estimation of allelic ratios.** Finally, we used cDNA-smMIPs to estimate allelic ratios, that is, allowing allele specific expression measurements. We designed 64 smMIPs for 32 common coding variants that are homozygous for the opposite allele in respectively the K562 and HEK293 cell line. We serially diluted K562 cDNA with HEK239T cDNA, so that the fraction of K562 cDNA decreased exponentially at a rate of 0.75. Because a target gene is not expressed at exactly the same level in the K562 and HEK293T cell line, one cannot a priori predict precisely the expected ratio for the first dilution step. However, between subsequent dilution steps the ratio of allelic ratios is expected to be 0.75. Indeed, this is what we observed (Supplementary Figs 12 and 13 and Supplementary Table 6). We then selected three single-nucleotide polymorphisms (SNPs) for each of which there were two smMIPs with non-overlapping extension and ligation probes, thus providing independent estimates. We found that the estimated allelic ratios were highly concordant between the two smMIPs (mean Pearson's $R^2 = 0.96$, Fig. 5).

## Discussion

cDNA-smMIPs have the potential to address an important practical need in gene expression studies as a method that can measure expression of a moderate number of genes of interest (∼10 to 500) across a significant number of conditions or samples (∼10 to 1,000 or more) at low cost. Based on our calculations, cDNA-smMIPs are more cost effective than qPCR, low-coverage RNA-Seq and CaptureSeq (Fig. 6, Supplementary Fig. 14 and Supplementary Table 7). cDNA-smMIPs allow multiplexing of hundreds of capture targets and at least 384 samples in a single sequencing run, using available barcoded PCR primers[8]. The high throughput of smMIPs has previously been

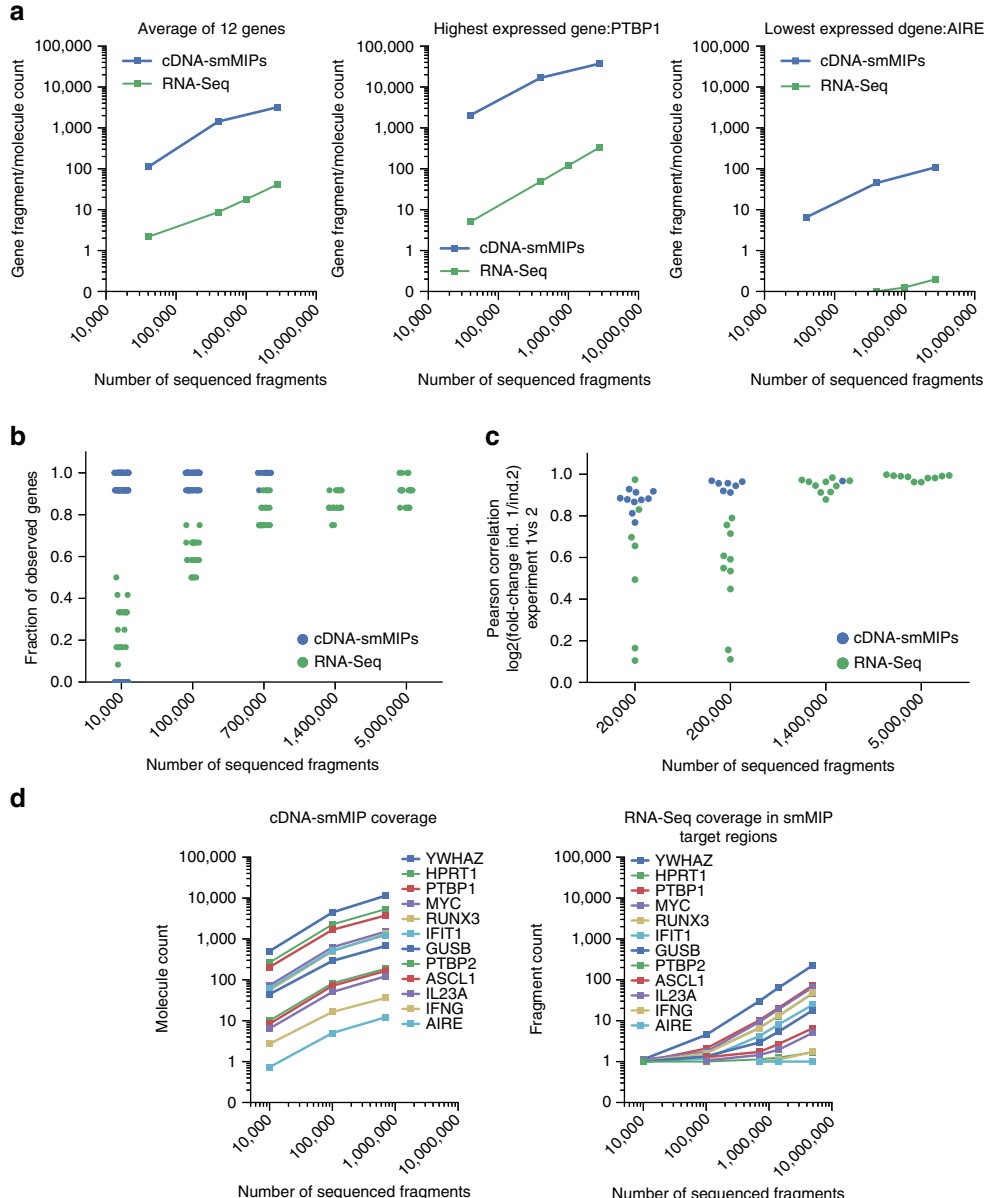

**Figure 3 | Comparison of cDNA-smMIPs with low-coverage RNA-Seq.** (**a**) Differences in molecule count (cDNA-smMIPs) and fragment count/read pairs (RNA-Seq) for the 12 genes targeted in the cDNA-smMIPs assay. The data points are averages over the randomly sampled sets of fragments. (**b**) Fraction of detected genes (genes with at least one mapped read) as a function of total number of reads. Each data point corresponds to a replicate. (**c**) Reproducibility of fold changes was estimated as a function of the total number of sequencing read pairs. For cDNA-smMIPs, the correlation is between log2(fold change) estimated in experiment 1 (using two technical replicates per individual) and experiment 2 (two technical replicates per individual). For the low-coverage RNA-Seq, correlation is between log2(fold change) estimated in experiment 1 (one technical replicate for respectively individual HG00117 and NA06986) and experiment 2 (one technical replicate for each individual). Each data point corresponds to a random sampling (without replacement) of the number of fragments (= read pairs) given on the horizontal axis and is based on 8 and 4 technical replicates for respectively cDNA-smMIPs and RNA-Seq. Corresponding scatter plots between the replicate DE estimates are shown in Supplementary Fig. 10. (**d**) Comparison of molecule counts (cDNA-smMIPs) and fragments/read pairs (RNA-Seq) mapping to the regions targeted by the cDNA-smMIPs. For each gene the average count across all smMIPs targeting the same gene is reported.

shown for DNA applications[8,17–19]. Total protocol duration is 2–3 days, with only ∼5 h of hands-on time. Furthermore, the protocol is highly amenable to automation. We have recently demonstrated such automation in the context of DNA-based resequencing with smMIPs[19].

To provide a reference for the performance of cDNA-smMIPs, we have included a comparison with CaptureSeq, a targeted RNA-Seq method based on an alternative enrichment strategy. However, it should be noted that CaptureSeq was primarily designed to target low-abundance RNA species, whereas our aim

was to estimate differential expression between many samples across the full dynamic range. We included in our panel of targeted transcripts a number of highly expressed genes (both for the ERCC standards and the endogeneous genes). These highly expressed targets still account for a substantial number of sequence reads; one can increase sensitivity to low-abundance transcripts by excluding highly expressed genes from the target panel.

There are several avenues for further improvement. First, we have not iteratively optimized the smMIP design or rebalanced

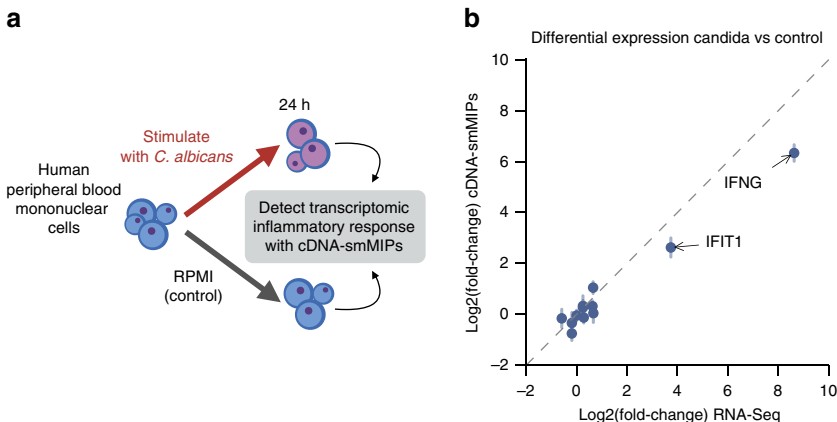

**Figure 4 | Expression changes following stimulation of PBMCs.** (**a**) Outline of PBMC stimulation experiment. (**b**) Concordance of differential gene expression (*Candida* versus Control) estimates from cDNA-smMIPs and previously published RNA-Seq data[16]. The cDNA-smMIPs and RNA-Seq experiments were performed on PBMCs from different individuals.

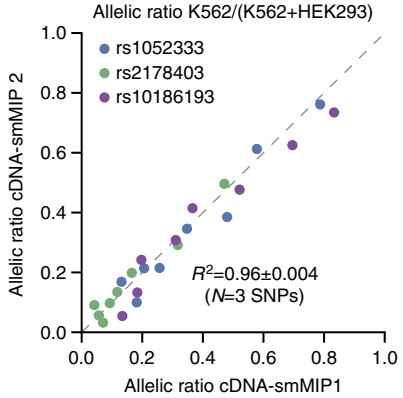

**Figure 5 | Estimation of allelic ratios with cDNA-smMIPs.** Concordance of allelic ratios estimated from distinct smMIPs (respectively non-overlapping extension probes and non-overlapping ligation probes) targeting the same SNP in a serial dilution of cDNA from K562 cell line with cDNA from HEK293 cell line (8 dilution steps). For all SNPs, the two cell lines are homozygous for the opposite allele. Reported variation in $R^2$ is s.d.

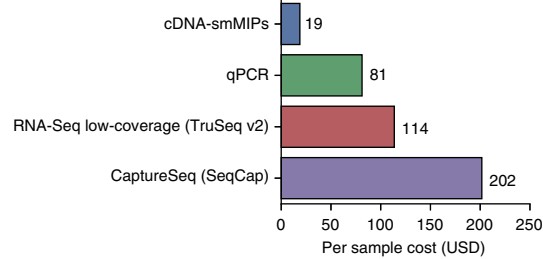

**Figure 6 | Cost comparison.** Reported cost is per sample, assuming a total of 1,000 samples, starting from RNA and including library preparation and sequencing. Breakdown of cost for cDNA-smMIPs is given in Supplementary Table 7. For cDNA-smMIPs and qPCR, calculation is based on 100 target regions in 20 genes; reported cost also includes cDNA synthesis (iScript), purification and measurement of cDNA concentration (8.25 USD). Cost for RNA-Seq is based on Illumina Truseq V2 kit (48 reactions, 3,724 USD, assuming 5 million paired-end reads on Illumina NextSeq at cost of USD 32.49). Cost for CaptureSeq is based on same Illumina Truseq V2 kit (48 reactions) followed by capture with Nimblegen SeqCap in 5-plex capture (87 USD/sample) as previously described[5,35] and 5 million paired-end reads on Illumina NextSeq(USD 32.49). The current commercially available version of SeqCap also permits 12-plex capture.

concentrations of smMIPs in the capture pool to even out variation in capture efficiency between smMIPs or expected transcript abundance. This is common practice for smMIP-based DNA-resequencing[8] and may also be beneficial for cDNA-smMIPs. Second, we find that the amount of input cDNA required (when not produced by ribosomal-depletion methods) varies with cell type, likely because the ratio between targeted transcripts and ribosomal RNA is variable. This may affect the number of unique molecules obtained from cDNA capture with smMIPs. Finally, the relatively low capture efficiency of smMIPs (compared with qPCR primers) increases the amount of input cDNA required to obtain a certain number of unique observations. This is reflected in the ratio of unique molecules to sequencing depth (Supplementary Tables 2–4). However, this is offset by the ability to count the individual molecules (transcripts) with smMIPs that strongly reduces the impact of PCR amplification and enables accurate quantification with modest amounts of input cDNA as we have shown here. The number of PCR cycles can generally be optimized to reduce PCR duplicates when samples are from the same tissue or cell type.

An important question is how to quantify the sensitivity of cDNA-smMIPs. Although cDNA-smMIPs can be applied to modest amounts of cDNA, our protocol cannot be directly applied to the extremely low amounts of cDNA typical of single-cell experiments. The probability to detect a given transcript furthermore depends on total sequencing reads and the other transcripts that are targeted in the same experiment. We therefore could not estimate from our data a sensitivity limit in terms of the minimum absolute number of target RNA molecules that must be present in a sample. It is however possible to characterize sensitivity in terms of the average number of transcripts per cell. Figure 2d shows that we detect transcripts that have an FPKM of <1 in corresponding whole-transcripts RNA-Seq data (Fig. 3d shows the absolute number of molecules detected with smMIPs for the corresponding gene, *AIRE*). Transcripts with an abundance of 10 fragments per kilobase of transcript per Million mapped reads (FPKM) in EBV cells on average correspond to 1 transcript copy per cell[20]. Thus, transcripts that have a low abundance relative to other transcripts in a cell can indeed be detected with cDNA-smMIPs. Furthermore, we expect that one can increase the probability of detecting extremely low-abundance transcripts by excluding high-abundance targets from the panel (because these reduce the probability that a low-abundance fragment will be sequenced) and increasing the amount of input cDNA.

We envision that cDNA-smMIPs may be useful to test the effect of many experimental perturbations (for example, drug treatments, overexpression of genes or CRISPR-Cas9 genome editing[21–23]) on the expression of genes in specific pathways. Another interesting application and direction for future research is whether cDNA-smMIPs can be applied to cDNA derived from degraded RNA such as from formalin-fixed, paraffin-embedded (FFPE) material. We have successfully applied smMIPs to DNA derived from FFPE material[19], but as RNA fragments may have degraded more extensively than DNA it is still an open question of whether cDNA from FFPE material is suitable for targeting with cDNA-smMIPs. A potential adaption of the approach described here could be to reduce the length of the target sequence of a smMIP (sequence between extension and ligation probe) so that potentially smaller cDNA fragments can be targeted with smMIPs.

In summary, we have shown that cDNA-smMIPs yield accurate estimates of differential expression, and are suitable to estimate relative gene expression and allele-specific expression. The use of molecular identifiers makes it possible to apply smMIPs to low quantities of input cDNA. The approach can be applied to cDNA from primary cells and cell line, is sufficiently scalable to perform hypothesis-driven expression analysis in large numbers of samples and is more cost effective than RNA-Seq and CaptureSeq.

## Methods

**EBV and PBMC smMIP design.** To design smMIPs for cDNA, we adapted a previously described workflow 'MipGen 1.0' for DNA[8,24]. This workflow uses a number of properties of a smMIP to predict its performance; these include properties of the MIP itself (for example, GC content) and its copy number in the human genome. The key difference with the design strategy for DNA is that we also design smMIPs that span exon–exon boundaries. It is customary to design qPCR primers that span an exon–exon boundary to improve specificity. However, we did not impose this requirement on the molecular inversion probes. Our script used the sequence of known transcripts from Ensembl (Version 75, hg19) as possible target sequences. We used the transcript that was estimated to be most highly expressed in EBV cell lines derived from European individuals from the Geuvadis project as the target sequence for which we designed probes. We converted the genomic coordinates of common variants from the 1000 Genomes Project (Phase I) to transcript coordinates. Together, the target sequences and known variants were used as input for the MipGen workflow. We did not use the tiling feature of MipGen. From the set of MIPs with a MipGen 1.0 performance score of 5, we manually selected 96 to approximately cover the transcript selected for each gene resulting in a mix of exon–exon crossing and within-exon smMIPs. smMIPs were designed with UMI of 9 nucleotides in between the extension probe sequence and the universal primer. The ligation and extension probe together were constrained to be 40 nucleotide length in total, each probe individually at least 16 nucleotide long and at most 24 nucleotides. With a common backbone sequence (that contains the universal primer binding sites) of 5′-CTTCAGCTTCCCGAT ATCCGACGGTAGTGT-3′, this resulted in smMIP oligos of $40+9+30=79$ nucleotides. The target sequence was designed to be 112 nucleotides. The designed cDNA-smMIPs are given in Supplementary Data 1.

**ERCC transcripts smMIP design.** We divided the ERCC transcripts into non-overlapping segments of 200 nucleotides. For 33 of 370 segments no smMIP with performance score of 5 could be found by MipGen 1.0, resulting in a total of 337 smMIPs for 92 ERCC transcripts. The median number of smMIPs per ERCC transcript was 4, with a range of 1–9. The designed cDNA-smMIPs are given in Supplementary Data 2.

**HEK293 K562 allelic ratio smMIP design.** In the absence of whole-genome sequencing data, we used public RNA-Sequencing data to call genotypes for common SNPs in HEK and K562 cell lines. We made the assumption that common variants in this data set would likely be present in the lines used in our laboratory. We aligned RNA sequencing data from study E-MTAB-3102 (https://www.ebi.ac.uk/arrayexpress/) for two HEK293 replicates (ERR688856 and ERR688857) and two K562 replicates (ERR688855 and ERR68885) to hg19 human genome reference with STAR version 2.4.2 (ref. 25). We used Samtools version 1.2 and bcftools[26,27] to call genotypes at all SNPs with allele count of at least 1,000 from release 0.3 of the Exome Aggregation Consortium (ExAC) database (142,659 sites in total). A set of stringent filters on differences genotype likelihood and coverage of both alleles was used to identify opposite homozygotes in the two cell lines. This resulted in a

set of 87 SNPs. To improve the comparability of the dilution curves for different SNPs, our goal to select genes that were expressed in both cell lines at similar levels. We selected the 32 SNPs for which the genes had the smallest difference in normalized coverage between the two cell lines. We first exhaustively generated all possible smMIPs covering these SNPs using the Ensembl (version 75) transcripts as targets. The probes were positioned such that in each transcript the targeted sequence was 100 bp. We used Burrows–Wheeler Alignment (BWA) tool to estimate the genome copy number count of each probe independently, and considered only smMIPs where both probes had genome copy count of 1. We then used version 2.0 of the MipGen software to calculate the predicted performance score. For each SNP we designed two smMIPs, covering as many annotated transcripts as possible while always taking the smMIP with the highest predicted performance score. All 64 selected smMIPs had predicted performance score of >0.50. The designed cDNA-smMIPs are given in Supplementary Data 3.

**smMIP analysis workflow.** Briefly, the workflow consists of the following steps:

1. Match each read pair to a designed smMIP probe (allowing two mismatches).
2. Remove likely extension–ligation dimers.
3. For all reads assigned to a given smMIP probe, identify the molecule counts from the unique molecular identifiers.
4. Determine the read-dependent threshold for UMIs due to sequencing errors.
5. Estimate normalized expression levels from error-corrected molecule counts integrating replicates using a Bayesian model.

For expression analysis, we determined molecule counts directly from the Fastq files. We did not first map the reads to a reference sequence to prevent mapping biases. Although a reference sequence was used to select the probe sequence of the smMIPs, which may introduce a bias, it is possible that the targeted sequence contains unannotated (small) exons.

**Correction for sequencing errors.** Sequencing errors in the UMI sequence of the smMIP may result in UMIs that are not associated with any cDNA molecule. As a result, after all UMIs associated with a molecule have been observed, the number of identified UMIs will continue to increase with sequencing depth until all possible UMIs have been generated by sequencing errors. This is especially problematic for low input samples where the number of unique molecules is low. With a sequencing error rate of 0.2% and a UMI of 9 nucleotides, the probability of at least one sequencing error is 1.8%. Thus, the error rate may account for a significant fraction of the UMIs.

We have chosen a simple but robust heuristic to remove UMIs due to sequencing errors. The idea is that sequencing error UMIs will tend to have low coverage. Thus, for a given smMIP, we sort all UMIs by their read coverage (that is, the number of reads with the same UMI) in descending order, and count only the UMIs for which the coverage is above a threshold $R$. $R$ is chosen such the UMIs with coverage greater than or equal to $R$ account for 95% of all the reads observed for the smMIP. This is effective both for complex libraries (where the majority of UMIs are observed only once) and libraries with many PCR duplicates.

**Bayesian hierarchical statistical model.** We constructed a statistical model to integrate observations from replicates into a single estimate of expression and to quantify uncertainty in estimates of differential expression. We used a negative binomial distribution to model the unique molecule counts. We defined expression for a given probe as the logarithm of the mean of the negative binomial distribution; this expression value is corrected for probe bias and normalized for sequencing depth. We assume that the overdispersion factor (relation between mean and variance) is the same for all probes. We allow for heterogeneity in the dispersion factor between experiments, as our results indicate that some experiments show more variability than others. The probe bias is estimated from differences in normalized counts (molecules per million molecules) between replicates and then used as a covariate in the model. We used Stan[28] to perform inference in this model using Markov chain Monte Carlo sampling. Stan was run independently for each condition (a condition is defined as a set of replicate experiments) to generate 1,000 independent samples. We used these samples from the posterior distribution to estimate differential expression between conditions. Details are given in the Supplementary Methods.

**Comparison with low-coverage whole-transcriptome RNA seq.** A number of samples in the GEUVADIS Project[15] were processed in replicate[29] in the same laboratory. We selected two technical replicates for individual HG00117 (1.M_111124_2 and 1.M_120209_1) and two technical replicates for individual NA06986 (1.M_111124_7 and 1.M_120209_1). These RNA-Seq libraries were all generated and sequenced at the University of Geneva. We subsampled the corresponding BAM files downloaded from ArrayExpress 10 times (without replacement) to total number of sequencing read pairs ranging from 200,000 to 5,000,000. We randomly sampled reads by first sorting on the read identifier and then selecting reads in contiguous blocks of the desired size. As a result, the subsampled BAMs also include unmapped reads, as would be expected in a low-coverage sequencing experiment. We used the individual downsampled BAMs

to estimate the number of fragments mapping to genes (using htseq-count), the number of detected genes (genes with mapped read count >0). We used BWA-MEM[30] to estimate the number of fragments mapping to the cDNA subsequences of 120 bp (Ensembl Version 75) targeted by the cDNA-smMIPs, as we found that BWA-MEM had higher sensitivity for identifying reads with partial matches than STAR. We determined reproducibility of differential expression estimates as follows: using DESeq[31], we estimated the log2(fold change) between samples HG00117.1.M_111124_2 and NA06986.1.M_111124_7 (experiment 1), and the log2(fold change) between samples HG00117.1.M_120209_1 and NA06986.1.M_120209_1 (experiment 2). We then computed correlation coefficients between the log2(fold change) estimates from experiments 1 and 2 across the 12 genes.

**Allelic ratio analysis.** For allelic ratio analysis, it was necessary to obtain allele counts for variants with a given reference sequence coordinate. As the UMI may affect mapability, we first processed the raw Fastq files to remove the UMI sequence from the read and to add the UMI to the read identifier using a custom script (Software). We then mapped these processed Fastq files to the reference sequence using STAR[25]. Next, we used a custom Python script to count the number of molecules covering each allele for the targeted SNPs, given the BAM file with mapped read pairs. Again, we first assigned each read to a smMIP probe by matching of the probe sequence to the known smMIP probe sequences (allowing two mismatches). We then proceed to analyse each target variant one by one. This is necessary as two different smMIPs may target the same SNP and sequence fragments with the same UMI but from different smMIPs should be considered as independent molecules. Then, for a given smMIP probe, we used a majority vote to call the allele for all reads with the same UMI.

**Allelic ratio dilution curve.** cDNA of K562 (ECACC) and HEK293T (ECACC) cells was combined in a serial dilution. cDNA of K562 and HEK293T cells was diluted to a final concentration of 1 ng μl[−1]. The serial dilution started with 75% K562 cDNA and 25% HEK293T cDNA. For the following steps of the serial dilution, 75% of the cDNA from the previous step was combined with 25% HEK293T cDNA. This resulted in a series of 8 cDNA samples of which the concentration K562 cDNA exponentially decreased and the HEK293T cDNA increased accordingly. One sample with only K562 cDNA and one sample with only HEK293T cDNA were also included in the experiment. Subsequently, samples were divided into two 10 μl duplicates containing 10 ng cDNA each. Capture was performed using the allelic ratio smMIPS.

The HEK cell line is listed in the ICLAC (International Cell Line Authentication Committee) database of commonly misidentified or contaminated cell lines. However, any possible contamination is irrelevant to this experiment as the dilution series provides an internal control.

**ERCC control transcripts.** The synthetic ERCC controls (Thermo Fisher Scientific, Waltham, MA, USA) were diluted 1:100, and 46 μl was used for cDNA synthesis after mixing with human PBMC total RNA using Superscript III Reverse Transcriptase (Thermo Fisher Scientific). Samples were purified using Qiaquick (Qiagen, Venlo, The Netherlands) columns, and cDNA was quantified using the Qubit ssDNA assay (Thermo Fisher Scientific). Of each sample 1 and 2 ng cDNA was used for MIP capture, and all capture experiments were performed in duplicate.

**Culturing of EBV cell lines.** EBV cell culturing was performed as described previously[32]. Briefly, human B-lymphoblast cells of two anonymized healthy individuals were immortalized by transformation with the Epstein–Barr virus[33]. Cells were cultured in RPMI-1640 medium (Sigma, St Louis, MO, USA) containing 10% (vol/vol) fetal bovine serum (Sigma), 1% 10 U μl[−1] penicillin and 10 μg μl[−1] streptomycin (Sigma), at a density of $0.5 \times 10^6$ cells per ml. Fresh medium was supplied twice a week. The anonymous EBV cell lines were obtained from the Radboud University Medical Center Human Genetics biobank. The use of the EBV cell lines is in accordance with the regulations of the Ethical Committee Arnhem-Nijmegen.

**Culturing of HEK cell line.** HEK293T cells (ECACC 12022001) were cultured in 100 mm tissue-culture treated culture dishes (Corning, New York, NY, USA) in Dulbecco's modified Eagle's medium containing 10% (vol/vol) fetal bovine serum, 1% penicillin/streptomycin and 1% sodium pyruvate (all from Sigma-Aldrich). At passage 12, cells were detached using 0.25% Trypsin (BD Biosciences, San Jose, CA, USA) after which ∼3 million cells were used for RNA isolation.

**Culturing of K562 cell line.** K562 cells (ECACC 89121407) were cultured in 75 cm[2] cell culture flasks (Corning) in RPMI-1640 medium containing 15% fetal bovine serum, 2% HEPES and 1% penicillin/streptomycin (all from Sigma-Aldrich). Approximately 6 million cells were used for RNA isolation.

**Mycoplasma contamination of cell lines.** All cell lines have been tested for mycoplasma contamination and were found to be negative.

**Stimulation of PBMCs.** Isolation and stimulation of PBMCs was performed as previously described[34]. In short, venous blood was collected into EDTA tubes and primary blood mononuclear cells were isolated by density centrifugation of blood diluted 1:1 in phosphate-buffered saline over Ficoll-Paque (Pharmacia Biotech AB, Uppsala, Sweden). Cells were washed three times in phosphate-buffered saline and resuspended in RPMI-1640 (Dutch modified) supplemented with 50 mg l[−1] gentamicin, 2 mM L-glutamine and 1 mM pyruvate. Cells were counted in a Coulter Counter Z (Beckman Coulter, Mijdrecht, The Netherlands) and adjusted to $5 \times 10^6$ cells per ml. Mononuclear cells ($5 \times 10^7$) in a 2 ml volume were added to round-bottom 6-well plates (Greiner, Alphen a/d Rijn, The Netherlands) and incubated with either culture medium (negative control) or HK *Candida albicans* ($1 \times 10^6$ per ml) for 24 h before isolation of RNA using RNeasy mini kit (Qiagen). PBMCs were isolated from buffy coats obtained after informed consent of healthy volunteers (Sanquin Bloodbank, Nijmegen, The Netherlands); this is approved by the Ethical Committee Arnhem-Nijmegen under no. CMO 2010-104.

**cDNA synthesis.** The isolated total RNA was quantified using the Qubit RNA HS assay kit (Thermo Fisher Scientific), cDNA synthesis of 2–5 μg of RNA was performed with iScript (BIO-RAD, Hercules, CA, USA) reverse trancriptase. After cDNA synthesis, the cDNA was purified using Qiaquick (Qiagen) purification columns, and cDNA quantity was measured using the Qubit ssDNA assay kit (Thermo Fisher Scientific).

**CaptureSeq.** We used the previously published data for the CaptureSeq protocol[5]. In that study, ERCC RNA was spiked into RNA from PBMCs from 9 human individuals (5 for ERCC1 and 4 for ERCC2); cDNA synthesis and enrichment using CaptureSeq was performed separately for each sample: RNA-Seq libraries were constructed with the TruSeq Stranded mRNA Sample Preparation Kit (Illumina) from RNA; each library was assigned to one of two multiplex capture pools, on which CaptureSeq enrichment was performed using SeqCap EZ oligonucleotides (Roche Nimblegen, Madison, WI, USA).

**smMIP capture.** The cDNA smMIP experimental procedure described below is largely based on the smMIP protocol developed for genomic DNA[8,13]. There are eight different steps involved in this experimental protocol: (1) smMIP pooling, (2) smMIP phosphorylation, (3) smMIP capture, (4) exonuclease treatment, (5) real-time PCR, (6) PCR, (7) sample pooling and purification and (8) Illumina Nextseq500 sequencing.

Regarding the smMIP pooling, three independent cDNA-smMIP pools were used for experiments testing differential expression of ERCCs (337 smMIPs), differential expression of EBVs and PBMCs (95 smMIPs) and allele-specific expression (64 smMIPs). In all experiments 5 μl of each of the 95 smMIPs were pooled into one single tube.

The concentration of the smMIP pool was calculated using the following relation: volume individual smMIP × concentration individual smMIP = volume smMIP pool × concentration smMIP pool. The used smMIP capture protocol was altered from the protocol for genomic DNA[8] that describes a MIP to gDNA molecule ratio of 800:1, corresponding to 264,000 MIP molecules per ng gDNA. For cDNA smMIPs we used 10-fold more smMIPs to compensate for the higher amount of RNA than DNA molecules per cell, that is, 2,640,000 smMIPs per ng cDNA. Per sample, 10 ng of cDNA in 10 μl (H2O) and for one blank (empty capture control) only 10 μl H2O was pipetted in a plate or strip tube. The cDNA input amount may differ per cell type, and for several samples we used 3 different input amounts, for example, 1, 10 and 50 ng.

For the smMIP phosphorylation an aliquot of 0.5 μl per smMIP, that is, one-tenth of the unphosphorylated pool, was used, resulting in 47.5 μl for 95 smMIPs. The mix for the phosphorylation (Supplementary Table 8) reaction contained: 1.9 μl T4 PNK (1 μl per 25 μl smMIPs, New England Biolabs, Ipswich, MA, USA), 6 μl 10 × T4 DNA ligase buffer with 10 mM ATP (New England Biolabs) and 4.6 μl H2O. This mix was transferred into PCR tubes, and placed in a thermocycler (DNA Engine, Bio-Rad, Hercules, CA, USA) to run the the smMIP phosphorylation program consisting of the following steps: (1) 37 °C for 45 min and (2) 65 °C for 2 min and (3) samples were cooled down to 4 °C (Supplementary Table 9).

The smMIP capture master mix was prepared for at least 30 reactions (due to low volume of required Ampligase DNA ligase). The capture master mix (Supplementary Table 10) for 30 reactions (450 μl) contained 75 μl 10 × Ampligase DNA ligase buffer (Epicentre/Illumina, Madison, WI, USA), 9.9 μl of the phosphorylated smMIP pool (0.833 μM diluted 1:625), 0.96 μl dNTPs (0.25 mM, diluted from 100 mM, Invitrogen/Thermo Fisher Scientific, Carlsbad, CA, USA), 9.6 μl Hemo Klentaq (10 units μl[−1], New England Biolabs), 0.30 μl Ampligase DNA ligase (100 units μl[−1], Epicentre/Illumina) and 354.3 μl H2O (added to get to total volume). Then, 15 μl of master mix was added to each cDNA sample. The lid offset temperature of the thermocycler was adapted to 10 °C, and the strip tubes were placed in a thermocycler. The capture program (Supplementary Table 11) contained the following steps: (1) 95 °C for 10 min to denature and (2) 60 °C for 24 h. At the end of the capture reaction, the samples were cooled on ice. The exonuclease treatment was performed immediately after the capture.

The exonuclease treatment master mix was prepared for the number of captured samples. Per sample, the exonuclease mix (Supplementary Table 12) contained 0.5 μl Exonuclease I (20,000 units ml$^{-1}$, New England Biolabs), 0.5 μl Exonuclease III (100,000 units ml$^{-1}$, New England Biolabs), 0.2 μl 10 × Ampligase DNA ligase buffer and 0.8 μl H$_2$O. The total volume of 2 μl mix was added to the captured samples, and the tube strips were placed back in the thermocycler. The exonuclease program (Supplementary Table 13) included the following steps: (1) 37 °C for 45 min and (2) 95 °C for 2 min and (3) samples were cooled down to 4 °C.

The real-time PCR (Rotorgene, Qiagen) was used to determine the amount of PCR cycles needed for sufficient amplification; this was variable among different cell types and different smMIP pools. Per sample the real-time PCR mastermix (Supplementary Table 14) contained 12.5 μl iProof HF Master mix (Bio-Rad), 0.125 μl Illumina PE FOR (100 μM forward primer, Supplementary Data 4), 0.125 μl Illumina PE BC1 (100 μM reverse primer, Supplementary Data 4), 0.125 μl SYBR green (10,000 × in dimethylsulphoxide, Thermo Fischer Scientific) and 7.125 μl H$_2$O. The total of 20 μl master mix was added to and mixed with 5 μl of exonuclease treated sample for the real-time PCR. The real-time PCR program (Supplementary Table 15) included: (1) 98 °C for 30 s, (2) 98 °C for 10 s, (3) 60 °C for 30 s, (4) 72 °C for 30 s, (5) return to step 2 for a total of 35 cycles, (6) 72 °C for 30 s and (7) cool down to 25 °C. Supplementary Fig. 15 shows an example image of the real-time PCR results; here, the estimated optimal number of PCR cycles was 21 cycles.

For the PCR reaction, 10 μl of each exonuclease-treated sample was pipetted into a strip tube and 1.25 μl of a different barcoded reverse primer (Supplementary Data 4) was added. The PCR mastermix (Supplementary Table 16) contained: 12.5 μl iProof HF Master mix (Bio-Rad), 0.125 μl Illumina PE FOR (100 μM forward primer, Supplementary Data 4) and 1.125 μl H$_2$O. The total of 13.75 μl master mix was added to and mixed with the 11.25 μl of combined sample and the barcoded reverse primer. The PCR program (Supplementary Table 17) included: (1) 98 °C for 30 s, (2) 98 °C for 10 s, (3) 60 °C for 30 s, (4) 72 °C for 30 s, (5) return to step 2 for a total of 21 cycles, (6) 72 °C for 30 s and (7) cool down to 4 °C. The PCR product was verified on agarose gel (Supplementary Fig. 16).

Before the Ampure XP bead purification, the bottle of AmpureXP beads (Beckman Coulter, Brea, CA, USA) was shaken thoroughly, since these beads settle overnight. The estimated volume to use was transferred to an Eppendorf tube, and let to adjust to room temperature for 30 min. Then, 70% ethanol was prepared freshly for each purification. An equal amount (5 μl) of each amplified sample was pooled into one Eppendorf tube, with a maximum of 96 samples per tube. The total volume of all samples in one tube was used to calculate the volume of Ampure XP beads to be added (0.85 μl Ampure XP beads per 1 μl sample). After adding the beads to the samples, a vortex was used to mix, and the mixture was incubated at room temperature for 10 min, and placed on a magnetic rack for 5 min. The beads were washed twice with 700 μl 70% ethanol; after the second wash, all ethanol was removed from the tube. The tube was left open to evaporate the residual ethanol and dry the beads. To elute the DNA, 25–50 μl (depending on amount of samples in the pool) of low TE was added to the beads and a vortex was used to resuspend all beads. Subsequently, the mixture was spun down, and placed on the magnetic rack for at least 1 min. The supernatant (now containing the DNA) was transferred to a new Eppendorf tube.

The final results of the smMIP capture were analysed on the Tapestation (Agilent Technologies, Santa Clara, CA, USA) (Supplementary Fig. 17) using the D1000 High sensitivity kit (Agilent Technologies). The final concentration of the smMIP-captured pool was measured in duplicate using the Qubit fluorometer (Thermo Fisher Scientific) and the Qubit dsDNA high sensitivity kit (Thermo Fischer Scientific). After quantification, the sample pool was diluted to 4 nM in 20 μl for sequencing on the Illumina Nextseq500.

Sequencing of smMIP libraries on the Illumina Nextseq500 requires spike in of custom primers. Therefore, 9 μl of custom primer 'MIPBC_SEQ_FOR' to cartridge position 20 (Read1), 9 μl of custom primer 'MIPBC_SEQ_REV' to cartridge position 21 (Read2) and 9 μl of custom primer 'MIPBC_SEQ_INDX' to cartridge position 22 (Index Read1) were added. The run was performed with 2 × 80 cycles, that is, 2 × 79 bp paired-end reads, and an 8 bp index read. Custom primer sequences as published previously were used (Supplementary Table 18, IDT, 100 μM, IDTE buffer).

All reagents used are specified in Supplementary Table 19; all equipment used is specified in Supplementary Table 20. The protocol is also described as a step-by-step procedure in the Supplementary Methods.

**Software.** Instructions, example of data sets and open-source software to design and analyse cDNA-smMIPs are available from https://github.com/keesalbers/cdna-smmips.

**Data availability.** All data are available in GEO under accession number GSE94800. This study used RNA sequencing data from study E-MTAB-3102 (https://www.ebi.ac.uk/arrayexpress/) for two HEK293 replicates (ERR688856 and ERR688857) and two K562 replicates (ERR688855 and ERR68885), as well as samples from the GEUVADIS Project[15]: two technical replicates for individual HG00117 (1.M_111124_2 and 1.M_120209_1) and two technical replicates for individual NA06986 (1.M_111124_7 and 1.M_120209_1). All other data are available from the authors on reasonable request.

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

## Acknowledgements

We thank Evan Boyle for the MIPGEN MIP performance score code and Klaas Mulder for comments. We thank Martin Jaeger and Theo Plantinga for assistance with the PBMC stimulation and culturing. C.A.A. received support from the FP7-PEOPLE-2013-IEF program of the European Commission (Grant Number 625095).

## Author contributions

C.A.A. and A.H. conceived the project. P.A., J.S., A.H. and C.A.A. designed the experiments. P.A., J.v.d.R., S.H.C.v.G. and M.S. performed experiments and analysed data. C.A.A. developed the statistical model and performed the main analyses. C.A.A. and P.A. drafted the manuscript. C.A.A. finalized the manuscript with contributions from all authors.

## Additional information

**Competing interests:** The authors declare no competing financial interests.

