## [Peer review file · Nature Communications]

REVIEWERS' COMMENTS:

Reviewer #2 (Remarks to the Author):

In their revised version, the authors have not yet addressed only my main comments.

The method proposed here is of certainly potential value, providing that the authors can demonstrate that it works efficiently on a range of biological samples. The use of scalable, targeted, sequencing methods is attractive and necessary, for instance for small biopsies but mostly for retrospective analyses exploiting archived material (e.g. FFPE blocks). The fact that the “DNA version” of this technology works on paraffin block does not warrant that cDNA-smMIPs does as well, and in my view, this should be at least tested, in order to delineate the method capabilities.

Another request was to estimate in which range of gene expression the technique is actually working reliably, in other words what is the limit of sensitivity of the technology presented. I appreciate that the authors now provide a comparison with low-coverage RNAseq, but notwithstanding this addition, the sensitivity of the proposed technique is still not clearly defined. It would be perhaps useful if the authors would report on the performance of cDNA-smMIPs in terms of number of RNA copies per cell. This would make it more interesting to a larger audience.

Point-by-point response to the reviewer comments

We thank the reviewer for the detailed comments which we feel have significantly improved our manuscript. Please find below a point-by-point response to all issues raised.

Reviewer's comment: It would be perhaps useful if the authors would report on the performance of cDNA-smMIPs in terms of number of RNA copies per cell. This would make it more interesting to a larger audience.

Author's reply: Even though cDNA-smMIPs can be applied to low amounts of cDNA (5 ng), we explicitly did not claim that cDNA-smMIPs can be applied to single-cell quantities (this is also not necessary for our technique to be of practical use) - and we clearly state this limitation now in the revised manuscript. Therefore we do not believe it is useful to quantify the limit of sensitivity in absolute terms of the number of transcripts of a single cell.

Nevertheless, cDNA-smMIPs allow detection of low abundant transcripts from bulk RNA as shown in Figure 2d and Figure 3d, and at increased sensitivity compared to whole-transcriptome sequencing at the same total number of sequence reads. As suggested by the reviewer, we can characterize sensitivity in terms of the average abundance of a transcript per cell, using previously published estimates of the relationship between FPKM and transcript abundance.

Furthermore, we believe that the most useful measure to determine applicability of our technique for potential users is the amount of input required to generate cDNA-smMIPs sequence libraries. These are modest, and we believe that most researchers will know for their particular application of interest how much total RNA they can isolate from the available number of cells and what the corresponding amount of cDNA will be. We have therefore reported the amounts of input cDNA we used to indicate the practical requirements for detection of low abundance transcripts with cDNA-smMIPs.

Finally, we note that we already stated in our manuscript that *"the relatively low capture efficiency of smMIPs (compared to qPCR primers) increases the amount of input cDNA required to obtain a certain number of unique observations."*

We believe this clearly states the limitations of smMIPs and the implications for required input material in light of limits on detection sensitivity. We estimate that smMIP capture efficiency is between 10% and 20%, but we feel that precise estimates of this number will not be of practical relevance as we already show which input amounts of cDNA result in successful sequence libraries.

To discuss the issue of sensitivity, we have added the following paragraph to the discussion:

"An important question is how to quantify the sensitivity of cDNA-smMIPs. Although cDNA-smMIPs can be applied to modest amounts of cDNA, our protocol cannot be directly applied to the extremely low amounts of cDNA typical of single-

cell experiments. The probability to detect a given transcript furthermore depends on total sequencing reads and the other transcripts that are targeted in the same experiment. We therefore could not estimate from our data a sensitivity limit in terms of the minimum absolute number of target RNA molecules that must be present in a sample. It is however possible to characterize sensitivity in terms of the average number of transcripts per cell. Fig. 2d shows that we detect transcripts that have an FPKM of <1 in corresponding whole-transcripts RNA-Seq data (Fig. 3d shows the absolute number of molecules detected with smMIPs for the corresponding gene, AIRE). Transcripts with an abundance of 10 FPKM in EBV cells on average correspond to 1 transcript copy per cell²⁰. Thus, transcripts that have a low abundance relative to other transcripts in a cell can indeed be detected with cDNA-smMIPs. Furthermore, we expect that one can increase the probability of detecting extremely low-abundance transcripts by excluding high-abundance targets from the panel (because these reduce the probability that a low-abundance fragment will be sequenced) and increasing the amount of input cDNA.”

Reviewer’s comment: The method proposed here is of certainly potential value, providing that the authors can demonstrate that it works efficiently on a range of biological samples. The use of scalable, targeted, sequencing methods is attractive and necessary, for instance for small biopsies but mostly for retrospective analyses exploiting archived material (e.g. FFPE blocks). The fact that the “DNA version” of this technology works on paraffin block does not warrant that cDNA-smMIPs does as well, and in my view, this should be at least tested, in order to delineate the method capabilities.

Author’s reply: We agree with the reviewer that application of cDNA-smMIPs to FFPE-derived cDNA is an important application that will be of interest to users of the proposed technique. We also fully appreciate that the demonstrated performance of smMIPs on FFPE-derived DNA is not a guarantee for application of smMIPs to FFPE-derived cDNA, as it is conceivable that the RNA from which the cDNA has been generated has degraded to a larger extent than the DNA. The true challenge may not be the smMIP capture but rather the efficient RT to cDNA. It is expected that FFPE-derived cDNA fragments are shorter than cDNA derived from non-degraded samples. In this light, we note that it is possible to reduce the length of the target sequence between the extension and the ligation probe of the smMIPs, so that in theory more of the smaller FFPE-cDNA fragments can be captured, the benefit of relatively short MIP capture-footprints was already proven beneficial for degraded FFPE DNA (<https://www.ncbi.nlm.nih.gov/pubmed/27637301>); in unpublished work we have used smaller insert smMIPs even on degraded DNA from toe nails and we have also further reduced the insert size to 54nt.

We feel that the required experiments to test this are not within the scope of this paper. However, we recognize the importance of the application and potential issues. We have added the following paragraph to the discussion:

“We envision that cDNA-smMIPs may be useful to test the effect of many experimental perturbations (e.g. drug treatments, overexpression of genes or CRISPR/Cas9 genome editing²¹⁻²³) on the expression of genes in specific pathways.

Another interesting application and direction for future research is whether cDNA-smMIPs can be applied to cDNA derived from degraded RNA such as from formalin-fixed paraffin-embedded (FFPE) material. We have successfully applied smMIPs to DNA derived from FFPE material¹⁹, but as RNA fragments may have degraded more extensively than DNA it is still an open question of whether cDNA from FFPE material is suitable for targeting with cDNA-smMIPs. A potential adaptation of the approach described here could be to reduce the length of the target sequence of a smMIP (sequence between extension and ligation probe) so that potentially smaller cDNA fragments can be targeted with smMIPs.